# Curiosity Driven Protein Sequence Generation via Reinforcement Learning

## ABSTRACT

Protein sequence design is a critical problem in the field of protein engineering. However, the search space for protein sequence design is incredibly vast and sparsely populated, which poses significant challenges. On the other hand, generative models struggle to adapt to different usage scenarios and objectives, leading to limited adaptability and generalization. To address these challenges, we explore a reinforcement learning algorithm based on latent space that enables protein sequence generation and mutation for different scenarios. Our approach has several advantages: (1) The reinforcement learning algorithm allows us to adjust the reward function according to different tasks and scenarios, enabling the model to generate and mutate protein sequences in a targeted manner. (2) The latent space mapped by ESM-2 is continuous, unlike the initial sparse and discrete space, and the curiosity mechanism further improves search efficiency. We evaluate our method in completely different scenarios, including different protein functions and sequences, and our experimental results demonstrate significant performance improvement over existing methods. We conduct multiple ablation studies to validate the rationality of our design.

## 1 INTRODUCTION

The essential mechanisms that maintain cellular health and vitality are mediated by proteins. Natural selection and repeated mutations on different proteins during biological evolution diversify characteristics, eventually amassing advantageous phenotype. Similar to this, directed evolution of proteins in protein engineering and design has shown to be a successful method for enhancing or changing the activities of the proteins or cellular fitness for industrial, research, and medicinal applications (Yang et al., 2018; Huang et al., 2016). Even with high-throughput screening from a diverse library, the protein sequence space of potential combinations of 20 amino acids is too big to be thoroughly searched in the lab (Huang et al., 2016). Alternatively, to phrase in another way, directed evolution has become trapped at local fitness maxima since library diversification is insufficient to get over fitness valleys and achieve nearby fitness peaks. In addition, functional sequences are uncommon and outnumbered by nonfunctional sequences in this enormous space of sequences.

In order to address the constraints, protein sequence design has been approached through the utilization of data-driven methodologies. The researchers employed a model-based approach utilizing reinforcement learning (RL) (Angermueller et al., 2020), as well as Bayesian optimization (Wu et al., 2017; Belanger et al., 2019) and generative models (Kumar & Levine, 2019; Jain et al., 2022; Hoffman et al., 2020). The protein functionality predictor was trained on experimental data to model the local landscape. Although significant progress has been made through the utilization of these techniques, the task of producing optimized sequences that have been experimentally confirmed remains a formidable challenge. It is suggested that the underlying reason for this phenomenon is twofold. One primary reason is that the optimization procedure typically involves the direct generation of candidate sequences through amino acid substitutions (Belanger et al., 2019) or additions (Angermueller et al., 2020). Due to the extensive search space, these methodologies exhibit significant computational inefficiency and frequently result in the investigation of regions within the space that possess a low probability of containing functional proteins. Prior research has investigated the optimization of a learned latent representation space for the purpose of designing biological sequences, as evidenced by the works of Gómez-Bombarelli et al. (2016); Stanton et al. (2022). This paper explores the optimization of sequences through the use of reinforcement learning (RL) in a latent representation

space, as opposed to the protein sequence space. Modifications made to the latent vector, such as small perturbations, can be interpreted as traversing a nearby functionality or fitness landscape within this representation space.

Though exploration in the representation space can be easier than in the original sequence space, the discrete action space can still be challenging to explore, i.e., the agent has 22 possible selections (including $N = 20$ amino acids and delimiter, terminator) in each position of a sequence. A notable category of methodologies depend on state novelty, wherein an inherent incentive in the guise of a 'novelty bonus' is granted on the basis of the frequency of state visitations (Sutton, 1990; Barto & Singh, 1991). Recent studies have expanded these methodologies to state spaces with high dimensions, where tabular counts are not feasible (Bellemare et al., 2016; Burda et al., 2018; Ostrovski et al., 2017). A distinct set of methodologies relies on artificial curiosity, wherein agents receive rewards commensurate with the predictive world model's prediction errors or information gains (Schmidhuber, 1991b; Bynagari & Amin, 2019). Methods based on curiosity have been expanded to accommodate more extensive state spaces.

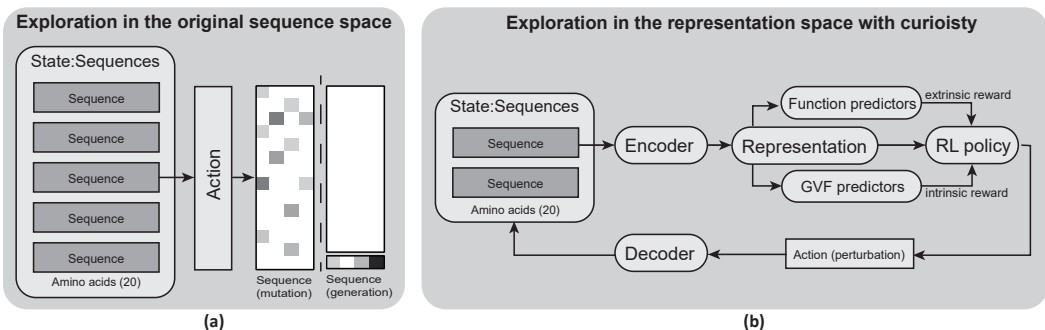

Figure 1: **The comparison of our method with directly search in the original protein space.** (a) Conduct a direct search within the raw protein sequence space utilizing an RL policy. (b) The primary framework of our approach. Initially, the protein space is mapped to a latent space through the utilization of an embedding model. Furthermore, the general value function (function predictor) is trained independently, utilizing the protein sequence's representation to furnish intrinsic rewards for the reinforcement learning (RL) agent. The RL policy is trained in a joint manner based on the intrinsic reward, which aims to encourage exploration, as provided by the GVF, and the extrinsic reward, which serves as the optimization goal and is provided by the environment.

In this paper, we explore a novel model that optimizes the protein sequence generation as an Markov Decision Process (MDP) in a representation space, and we further explore a curiosity driven mechanism to encourage the agent's exploration with higher efficiency. We demonstrate the main structure of our method in Fig. 1. Our method trains an RL policy that learns to generate/mutate the protein sequences. At each timestep, the policy updates the latent representation by small perturbations to maximize proteins functionality. On the other hand, the policy receives the intrinsic rewards that provided by general value functions (GVFs) (Sutton et al., 2011) to pose questions about the cumulative future value of state-dependent features. More specifically, we train an ensemble of predictors to minimize the TD-errors of the GVFs and derive an intrinsic reward based on the TD-errors and disagreements in the long-term predictions. We evaluate our method in two evaluation tasks, optimizing the functionality of the green fluorescence protein (GFP) and the cell fitness of imidazoleglycerol-phosphate dehydratase (His3). Our results show that the proposed framework design sequences with higher protein functionality and cellular fitness than existing methods. Ablation studies show that, based on the evaluation of various model options for state and action for the RL framework, the proposed latent representation update can successfully optimize the protein and search the vast design space.

## 2 RELATED WORKS

**Generative model** The protein sequence generation and has been studied in numerous generative models (Shi et al., 2020; Luo et al., 2021), MCMC methods (Grathwohl et al., 2021; Yang et al., 2018; Seff et al., 2019; Xie et al., 2021), RL (Segler et al., 2017; Cao & Kipf, 2018; Popova et al., 2019; Gottipati et al., 2020) and evolutionary methods (Brown et al., 2004; Swersky et al., 2020). Some of

these methods rely on a given set of "positive examples" (high-reward) to train a generative model, thus not taking advantage of the "negative examples" and the continuous nature of the measurements (some examples should be generated more often than others). Others rely on the traditional return maximization objectives of RL, which tends to focus on one or a few dominant modes.

**Protein Language Model** Techniques such as BERT (Devlin et al., 2019)and GPT-2 (Radford et al., 2019) have been utilized to train protein language models due to their similarity to tasks in natural language processing (NLP). The protein language model (Alley et al., 2019; Brandes et al., 2021; Ferruz et al., 2022), was trained on a dataset comprising 250 million sequences. The model has exhibited proficiency in generating representations that effectively capture protein structure and communicate significant biological characteristics. The study demonstrated that generalization of the learnt representation across diverse applications is crucial for achieving state-of-the-art outcomes in supervised prediction of mutational effects. Representational learning has also been employed in the study of protein structure.

**Exploration** The maintenance of explicit counts for states or state-action pairs is a technique employed in tabular reinforcement learning to facilitate exploration and attain effective learning (Strehl & Littman, 2008; Jaksch et al., 2008). Density models have been utilized to extend count-based bonuses to larger state spaces by generating pseudo-counts (Ostrovski et al., 2017; Bellemare et al., 2016). Novelty bonuses based on estimated novelty of the visited state are heuristics that provide a bonus, as inspired by State novelty (Burda et al., 2018). One approach to incorporating exploration via curiosity involves providing incentives for a policy to encounter state transitions that are unexpected by a learning predictor (Schmidhuber, 1991b). This approach is appropriate for deterministic settings, but it is susceptible to the "noisy TV problem" when applied to stochastic environments. In order to address this constraint, intrinsic rewards have been delineated based on learning progress (i.e., the primary derivative of the prediction error) (Schmidhuber, 1991a), information gain (Bynagari & Amin, 2019), or compression progress (Schmidhuber, 2007). Additionally, it is worth comparing these bonuses to pseudo-counts (Bellemare et al., 2016) as described in reference. An alternative approach to reducing sensitivity to noise involves the assessment of prediction errors in a latent space (Pathak et al., 2017). The approach we employ differs from the state count and novelty methods in that it incorporates temporally extended values beyond the present observation into its model. Incorporated within the framework is a measure of state-novelty, denoted as RND (Sutton et al., 2011), which is treated as a distinct scenario in instances where the GVF discount factors assume a value of zero.

## 3 PROBLEM FORMULATIONS

**Reinforcement Learning.** We formulate a single sequence $x$ design as a Markov Decision Process (MDP) with $\mathcal{M} = \langle \mathcal{S}, \mathcal{A}, \mathcal{R}, \mathcal{T} \rangle$. Specifically, $\mathcal{S}$ denotes the set of all possible sequences, i.e., $\mathcal{S} = \cup_{t=0,\cdots,T} seq_t$. $\mathcal{A}$ indicates the set of actions that modifies the sequence. At each time-step $t$, the agent takes an action $a_t \in \mathcal{A}$ that modifies the sequence $s_t$ to $s_{t+1}$ according to the transition dynamics $P(s_{t+1}|s_t, a_t) = 1$. The reward function $R(s_t, a_t)$ is expected to be zero except when $t = T$, where $T$ is the horizon of each episode. The goal is to learn the control policy $\pi^* : \mathcal{S} \to \mathcal{A}$ mapping states to actions that maximizes the expected return $\mathbb{E}[\mathcal{R}_t]$, which takes into account reward at future time-steps $J(\pi) = \mathbb{E}[\mathcal{R}_t] = \mathbb{E}\left[\sum_{t=0}^{T} \gamma^t r_t\right]$ with a discount factor $\gamma \in [0, 1]$. In this paper, we define the reward function as $r(s_t, a_t) = r_e(s_t, a_t) + r_i(o_t, a_t)$, where $r_e(s_t, a_t)$ indicates the extrinsic reward provided by the environment (results of the protein function evaluation), which can be regarded as the optimization objective of the protein sequence. $r_i(o_t, a_t)$ denotes the intrinsic reward provided by the general value function, which aims at encouraging the agent's exploration. Now we introduce more details of the intrinsic reward and general value function.

**General Value Function (GVF).** The GVF in our method is trained to predict the uncertainty of the RL agent in state space, and it requires exploration data. In order to circumvent this cold-start issue, we introduce a random network distillation (RND) (Sutton et al., 2011) that provides the agent a state novelty reward proportional to the error of making predictions $\hat{z}(s_t)$, where the targets are generated by a randomly initialized network $Z_\phi : \mathcal{O} \to \mathbb{R}^d$. Then, the RND intrinsic reward for an observation is given by

$$r_i(o_i) = ||Z_\phi(o_t) - \hat{z}(o_t)||_2, \qquad (1)$$

RND-like error signals can be a simple way to obtain effective uncertainty estimates about targets at a given input, which could subsequently be used for exploration in RL or detecting out of distribution samples. Based on RND, we further define a general value function (GVF) based on a policy $\pi$ and a pseudo reward function $Z : \mathcal{O} \to \mathbb{R}$. The GVF is defined as

$$v_{\pi,z} = \mathbb{E}_\pi \left[ \sum_{k=0}^{\infty} \gamma_z^k Z(O_{t+k}) | O_t = o \right], \tag{2}$$

General value functions extend the concept of predicting expected cumulative values to arbitrary signals beyond the reward. Predictions from GVFs have previously been used as features for state representation or to specify auxiliary tasks that assist in shaping representations for the main task.

## 4 METHODOLOGY

### 4.1 SEQUENCE REPRESENTATION

We first denote $x \in \mathcal{X}$ as a protein sequence of length $L$, where $L$ indicates the number of amino acids in the sequence and $\mathcal{X}$ is the sequence space. Given an $x$, we employ the ESM-2 model (Lin et al., 2023) to encode it to a representation $y$, which is a protien language model pre-trained on the Uniref50 dataset (Suzek et al., 2014). This model uses a BERT (Devlin et al., 2019) encoder and is trained with 150 million parameters in an unsupervised fashion. ESM-2 is utilized to map mutation effects onto a latent space in our model. Given the sequence $x$ as an input, ESM-2 outputs a matrix $m \in M$ of dimensions $(L + 2, E)$, where $E$ is the dimension size of the embeddings, and $M$ is the embedding space. In ESM-2, a CLS (classification) token is prepended to the sequence, and an EOS (end-of-sequence) token is appended to the sequence when generating the embeddings. They serve as sequence-beginning and sequence-ending characters, respectively.

Prior to being sent to the policy, the output of the encoder with dimensions $(L+2, E)$ is pre-processed. In order to reduce the dimension from $(L + 2, E)$ to $(E)$, we only employ the CLS token embedding. Nevertheless, given that even the lowest ESM-2 model's embedding size E is 480, this might still be a sizable amount for the action space. Therefore, we further reduce the dimension to $(R, )$ and produce the final representation using 1-dimensional adaptive average pooling.

The amino acid sequence $x$ can be recovered from a reduced representation $y$, which has a size of $(R, )$, through the utilization of the sequence decoder. A linear layer is utilized to increase the representation size to $(E, )$. In order to restore the initial embeddings' dimensions of $(L + 2, E)$ that were derived from ESM-2, the reduced representation's dimensions are initially expanded. Subsequently, the aforementioned matrix is merged with the wild-type embeddings that were acquired through the utilization of ESM-2, which is a pre-existing encoder of dimensions $(L + 2, 2E)$, and is succeeded by a linear layer. Subsequent to traversing a dropout layer, the restored representation is conveyed to the language model head of the decoder. This component is responsible for remapping the representation to the sequence space, thereby predicting the amino acid sequence from its corresponding embeddings. The resultant sequence $x$ denotes the retrieved output.

### 4.2 A GENERAL VALUE FUNCTION TO PREDICT THE FUTURE

ESM-2 is utilized to facilitate the mapping of the primary protein sequences to a latent space, thereby mitigating the intricacy of the search process. However, it is noteworthy that the action space remains discrete and intricate, necessitating the selection of each action from a pool of $N = 20$ feasible actions. Consequently, drawing from the latent space, we have devised a curiosity mechanism that is grounded in the value function. This mechanism serves to furnish intrinsic rewards, which rewards the agent for taking actions that lead to higher uncertainty about the future. In order to streamline this process, we train a separate neural network to predict the pseudo-values based on the current observations and actions. We denote this predictor as $\hat{v}_{\pi,z} : \mathcal{H} \to \mathbb{R}^d$, where $z$ is a state novelty reward proportional to the error of making predictions.

**General value function.** The General Value Function (GVF) is utilized for the purpose of forecasting outcomes that are extended over time within a given environment. At a given time step $t$, the present observation $o_t$ is associated with a set of pseudo-rewards $z_{t+1} \in \mathbb{R}^d$. These rewards, in conjunction with the policy, are utilized to determine the anticipated total of pseudo-rewards in the future. Prior

research has indicated that the arbitrary characteristics extracted by a neural network are frequently satisfactory in capturing advantageous attributes of the input (Koan, 2008). Likewise, it is possible to utilize neural network structures to articulate pre-existing knowledge regarding advantageous features (Ulyanov et al., 2017). In order to achieve this objective, we derive simulated rewards from a neural network $Z_\phi \to \mathbb{R}^d$ that is fixed and initialized randomly, with parameters $\phi$ that enable it to map observations to simulated rewards. This option also circumvents the challenging issue of identifying significant general value functions, as stated in the reference cited.

**How to train the GVF predictors?** We train a separate neural network to predict the pseudo-reward that encourage the agent for taking acitons that generate previously unknown outcomes. More specifically, we train a predictor $\hat{v}_{\pi,z} : \mathcal{H} \to \mathbb{R}^d$ maps histories to pseudo-rewards. The predictor is trained on-policy, implying that the GVFs are evaluated under the current policy. One motivating factor for this choice is that it couples the prediction task to the current policy, thus creating an incentive to vary behaviour for additional exploration Flet-Berliac et al. (2021). We use the $\lambda-$return as the target for the predictor, which can be recursively expressed as:

$$G_t^z(\lambda_z) = Z_{t+1} + \gamma_z(1 - \lambda_z)\hat{v}_{\pi,z}(H_{t+1}) + \gamma_z\lambda_z G_{t+1}^z(\lambda_z), \tag{3}$$

where, $\lambda_z \in [0, 1]$ is the parameter that allows balancing the bias-variance trade off by interpolating between TD(0) and Monte Carlo estimates of the pseudo-return . This predictor is trained to minimize the mean squared TD-error with the $\lambda-$ return target. We denote $G_t^z(\lambda_z)$ as $G_t^z$ for convenience.

## 4.3 EXPLORATION WITH THE DISAGREEMENT ON GVF

To generate an intrinsic reward, a straightforward choice could be to consider the error between the temporal difference target (in Eq. 3)and the predictor's output at the current observation:

$$L_{TD}(h_t) = [G_t^z - \hat{v}_{\pi,z}(h_t)]^2, \tag{4}$$

However, a challenge that results from broadening the task's scope is the presence of aleatoric uncertainty. The agent should concentrate on the reducible epistemic uncertainty from an exploratory standpoint rather than the irreducible aleatoric uncertainty (Kiureghian & Ditlevsen, 2009). If the predictor is used to directly minimize Eq. 4, it would overlook the intrinsic variance in the TD-target, which is caused by the stochasticity in the policy and environment. Consequently, utilizing the prediction error of the GVF target as an intrinsic reward would fail to differentiate between aleatoric and epistemic uncertainty.

In order to address the challenge posed by aleatoric uncertainty in GVF, we adopt the approach suggested in Jain et al. (2022) whereby we train a set of predictors and leverage the variability observed across their predictions as a means of multiplying the prediction error. To be more precise, a set of $K$ predictors denoted as $\hat{v}_{\pi,z}^k$ with $k$ belonging to the set $\{1, 2, \cdots, K\}$ are trained. Each member of the ensemble has the same prediction target, which is the $\lambda-$pseudo-return (as shown in Eq. 4). This is achieved by utilizing bootstrapped values from the predictions made by that particular member. The intrinsic reward is determined through the utilization of a collection of predictors.

$$R_i(o_t) = \sum_{j=1}^{d} \left( \mathbb{E}\left[ L_{\text{TD}}^k(h_t) \right] \odot \mathbb{V}\left[ \hat{v}_{\pi,z_j}^k(h_t) \right] \right)_j \tag{5}$$

$$= \sum_{j=1}^{d} \left[ \frac{1}{K} \sum_{k=1}^{K} \left( G_t^{z_j} - \hat{v}_{\pi,z_j}^k(h_t) \right)^2 \right] \cdot \left[ \frac{1}{K-1} \sum_{k=1}^{K} \left( \bar{v}_{\pi,z_j}(h_t) - \hat{v}_{\pi,z_j}^k(h_t) \right)^2 \right]$$

where $\odot$ corresponds to element-wise multiplication. In this formulation, even when prediction error remains, the exploration bonus will vanish as the predictors converge to the same expectation.

## 5 EXPERIMENTS

### 5.1 EXPERIMENTS SETUP

Proteins with varying lengths and functions were selected to ensure a comprehensive assessment. Specifically, the green fluorescent protein (GFP), which occurs naturally in jellyfish, and imidazoleglycerol phosphate dehydratase (His3), a crucial enzyme in the human body, were chosen for

their robustness in evaluation. The dataset (Sarkisyan et al., 2016) is utilized for the training of the GFP encoder-decoder and its corresponding functionality predictor. The dataset comprises a total of 54,025 mutant sequences, each of which is associated with a corresponding log-fluorescence intensity value. The protein sequences have a length denoted by $L$, which is equal to 237. The dataset utilized for the His3 protein pertains to its evolutionarily-relevant segments and their corresponding mutant sequences, as well as their associated growth rate, as reported in the study by Pokusaeva et al. (2019). The data underwent processing resulting in 50,000 sequences with a length of $L = 30$. The aforementioned dataset is utilized for the purpose of training the His3 encoder-decoder and fitness predictor. The datasets are partitioned into distinct training and testing sets with a ratio of $90 : 10$, without any overlap between them. The training set mentioned earlier is utilized by both the encoder and the decoder. The utilization of the test set is primarily for the purpose of assessing the performance of the trained models and determining the optimal starting points for the process of optimization.

**Implementation Details** The utilization of a pre-trained ESM-2 model (Lin et al., 2023) with 150 million trainable parameters is employed by both the encoder and decoder,. Distinct ESM-2 models are utilized to train optimization and evaluation oracles. The optimization oracle is trained utilizing a model consisting of 150 million parameters, whereas the evaluation oracle employs a model with 35 million parameters. The dimensionality of the latent representation space has been specified as $R = 8$, thereby determining the dimensions of both the state and action vectors. The perturbation magnitude denoted by $\epsilon$ is uniformly assigned a value of $0.1$ for each element of the action vector. The duration of the episode denoted by $T$ has been established as 20.

**Evaluation Metrics** We use three evaluation metrics as reported in Jain et al. (2022): performance, novelty, and diversity. We also consider two additional metrics for robustness: the originality of optimized sequences, named as original, i.e. sequences not contained in the training dataset, and the distance between the optimized sequence and wild-type, named as dist(WT). The performance evaluation metric is calculated as the mean predicted functionality from the top $K$ generated sequences. Let the generated sequences be contained in the following set $\mathcal{G}^* = \{g_1^*, \cdots, g_K^*\}$, performance is defined as $1/K \sum_i f(g_i^*)$. The novelty evaluation metric is defined to assess if the policy is generating sequences similar to the ones contained in the experimental data. Defining $\mathcal{P}$ as the experimental data set containing the wild-type protein sequence, novelty is given as follows:

$$\frac{1}{K \cdot |\mathcal{P}|} \sum_{g_i^* \in \mathcal{G}^*} \sum_{p_j \in \mathcal{P}} dist(g_i^*, p_j), \tag{6}$$

where $dist$ is defined as the number of different amino acids of two sequences. The diversity evaluation metric is defined as the mean of the number of amino acids that are different among the optimized sequences, and can be expressed as:

$$\frac{1}{K(K-1)} \sum_{g_i^* \in \mathcal{G}^*} \sum_{g_j^* \in \mathcal{G}^* - \{g_i^*\}} \text{dist}\left(g_i^*, g_j^*\right) \tag{7}$$

The original metric is defined as $1/K \sum_i 1([g_I^* \notin \mathcal{P}])$ and the distance from wild-type (WT) metric is given as $1/K \sum_{g_i^* \in \mathcal{G}^*} \text{dist}(w, g_i^*)$, where $w$ is the wild-type sequence. When testing the protein functionality of GFP, we include the presence of the chromophore region (residues SYG in the wild-type protein) in the optimized sequence, as these residues are related to the ability to emit fluorescence.

## 5.2 MAIN RESULTS

In each method, we totally generate 160 sequences (20 random seeds and 8 sequence for each seed), and evaluate the 10 highest-performing. We have two types of tasks (generation, mutation) in each dataset.

**GFP Sequence Generation.** We report the evaluation results for GFP sequence generation in Table. 1 and GFP sequence mutation in Table. 2. In this study, we conducted a comparison of four optimization techniques, i.e., PEX Ren et al. (2022), BO (bayesian optimization) (Swersky et al., 2020), CbAS (adaptive sampling) (Brookes et al., 2019), GFlowNet (generative model) (Jain et al., 2022), and DynaPPO (reinforcement learning) (Angermueller et al., 2020). Note that we follow the original

settings of PEX in mutation tasks, and "PEX (radius =10)" is the optimal performance. in mutation tasks and The phrase "directed evolution" pertains to the mean functional values of the starting states employed in the process of optimization. In addition, we incorporate stochastic variations in the initial states. Table. 1 demonstrates that the proposed method exhibits superior performance compared to directed evolution. Among the considered techniques, solely the proposed method and CbAS exhibit efficient optimization of GFP, while BO, GFlowNet, and DynaPPO demonstrate suboptimal performance. Despite the fact that our approach restricts the operation to a minor increment in the latent space, it exhibits greater levels of originality and variety in comparison to CbAS, which similarly constrains the exploration of the search space during optimization. The findings of this study are noteworthy as they indicate that two of the GFP sequences that underwent optimization through the proposed method exhibited greater predicted functionality, in comparison to the experimental wild-type functionality. It is noteworthy to observe that the methods that attain the maximum deviation from the wild-type sequence are the ones that exhibit the minimum level of performance.

| Model | Performance | Novelty | Original | dist(WT) | Diversity |
|---|---|---|---|---|---|
| **Ours** | $\mathbf{3.893 \pm 0.413}$ | 8.232 | 100% | 8.100 | 7.422 |
| PEX | $3.135 \pm 0.322$ | 9.154 | 100 % | 10.021 | 6.849 |
| BO | $0.565 \pm 0.095$ | 30.18 | 100% | 32.70 | 6.567 |
| CbAS | $2.304 \pm 0.178$ | 7.361 | 70% | 6.682 | 2.047 |
| Directed evolution | $2.687 \pm 0.237$ | 7.704 | - | 6.849 | 4.858 |
| DynaPPO | $0.004 \pm 0.003$ | 218.9 | 100% | 219.3 | 224.1 |
| GFlowNet | $0.003 \pm 0.002$ | 199.4 | 100% | 200.1 | 12.53 |

Table 1: **Test performance of the generation tasks in GFP.** Based on the obtained outcomes, it can be concluded that our approach exhibits a significantly superior performance compared to the baseline method. Our approach exhibits clear benefits in terms of sequence diversity while also ensuring optimal performance.

| Model | Performance | Novelty | Original | dist(WT) | Diversity |
|---|---|---|---|---|---|
| **Ours** | $\mathbf{5.414 \pm 0.710}$ | 6.891 | 100 % | 10.42 | 9.102 |
| Random - 1 | $2.622 \pm 0.113$ | 6.611 | 80 % | 7.470 | 6.905 |
| Random - 5 | $2.930 \pm 0.250$ | 13.710 | 100 % | 10.55 | 13.07 |
| PEX (radius =10) | $4.250 \pm 0.196$ | 10.721 | 100% | 8.026 | 12.425 |
| BO | $1.815 \pm 0.290$ | 24.87 | 100% | 32.70 | 8.030 |
| CbAS | $1.606 \pm 0.208$ | 8.620 | 80% | 7.120 | 2.301 |
| Directed evolution | $3.657 \pm 0.337$ | 4.704 | - | 5.019 | 5.011 |
| DynaPPO | $0.006 \pm 0.002$ | 143.2 | 100% | 142.3 | 180.4 |
| GFlowNet | $0.003 \pm 0.001$ | 121.4 | 100 % | 210.9 | 20.45 |

Table 2: **Test performance of the mutation tasks in GFP.** The findings indicate that the approach employed can considerably enhance the efficacy of protein testing via genetic alterations. Random-1 and Random-5 denote stochastic alterations at one and five specific nucleotide positions, respectively, which aligns with the conventional approach utilized in the majority of protein engineering.

**His3 Sequence Generation** We report the evaluation performance of our method with baseline methods in Table. 3 (generation tasks) and Table. 4 (mutation tasks), wherein our framework demonstrates superior performance. The optimization of His3 was found to be ineffective when utilizing DynaPPO and BO. It is noteworthy to observe that a solitary arbitrary mutation is a viable strategy, considering the His3 protein's mere 30-unit length. It should be noted that while the aforementioned approach attains superior overall performance, its level of originality is comparatively lower than that of all alternative methods. This suggests that the decoder is able to retrieve comparable sequences within the space of representation. In comparison to CbAS, the method under consideration attains greater levels of novelty and diversity.

## 5.3 ABLATION STUDIES

**Representation space** The present study conducted a controlled experiment to investigate the necessity and suitability of employing ESM-2 for the embedding of raw protein sequences. Consequently, we devised three distinct control groups. The first group involved no embedding, whereby we explored and trained within the original protein sequence space. The second and third groups utilized ESM-1v (Meier et al., 2021) and Ablang (Olsen et al., 2022), respectively, to embed protein sequences into the latent space. The control experiment's dataset comprises GFP, and the outcomes

| Model | Performance | Novelty | Original | dist(WT) | Diversity |
|---|---|---|---|---|---|
| **Ours** | **0.838 ± 0.108** | 8.723 | 90% | 9.570 | 9.627 |
| PEX | 0.810 ± 0.112 | 8.223 | 100 % | 8.950 | 9.358 |
| BO | 0.202 ± 0.065 | 26.11 | 100 % | 28.50 | 4.577 |
| CbAS | 0.552 ± 0.120 | 8.280 | 90% | 5.210 | 2.690 |
| Directed evolution | 0.544 ± 0.092 | 5.313 | - | 5.410 | 4.392 |
| DynaPPO | −0.120 ± 0.102 | 23.01 | 100% | 21.45 | 25.70 |
| GFlowNet | 0.080 ± 0.004 | 24.80 | 100% | 26.47 | 27.82 |

Table 3: **Test performance of the generation tasks in His-3.** Despite the comparatively shorter length of the protein in His3 as compared to GFP, our approach exhibits a significant improvement over the baseline methods. Moreover, our method offers distinct advantages in terms of the diversity of the generated protein sequence, thereby demonstrating its potential for efficient exploration of the protein sequence space.

| Model | Performance | Novelty | Original | dist(WT) | Diversity |
|---|---|---|---|---|---|
| **Ours** | **0.979 ± 0.098** | 7.306 | 70% | 10.27 | 3.325 |
| Random-1 | 0.835 ± 0.060 | 7.032 | 80% | 7.660 | 7.724 |
| Random-5 | 0.698 ± 0.090 | 10.01 | 100% | 9.635 | 14.27 |
| PEX (radius=10) | 0.956 ± 0.104 | 8.541 | 100 % | 8.230 | 9.424 |
| BO | 0.202 ± 0.065 | 26.11 | 100 % | 28.50 | 4.577 |
| CbAS | 0.749 ± 0.157 | 7.287 | 90% | 4.700 | 2.356 |
| Directed evolution | 0.616 ± 0.110 | 6.889 | - | 6.710 | 6.942 |
| DynaPPO | −0.201 ± 0.142 | 27.41 | 100% | 26.70 | 27.47 |

Table 4: **Test performance of the mutation tasks in His3.** The His3 dataset exhibits a diminutive average length of protein sequences. Thus, despite the absence of a substantial enhancement in performance relative to random mutation, our approach outperforms random mutation in terms of overall performance, novelty, and diversity, while exhibiting a high degree of stability.

of the control experiment are presented in Table. 5. The findings demonstrate that the utilization of the embedding model to facilitate the exploration of the RL agent in the latent space can significantly enhance the learning efficacy and ultimate performance. This outcome is in line with our earlier analysis, which underscores the crucial role of the discrete protein sequence space in the RL agent's exploration. The acquisition of language can prove to be a formidable task, and its complexity may hinder the provision of constructive criticism in a training setting. ESM-2 is utilized for the purpose of embedding due to its extensive pre-training data set, which enables it to more effectively conform to protein sequences across various scenarios. Despite the fact that the Ablang model has a lower number of parameters compared to the ESM-2 model, there is a negligible discrepancy in the training cost between the two models. Specifically, the ESM-2 model necessitates approximately 140 cpu hours, while the Ablang model requires approximately 122 cpu hours. Based on the observed discrepancy in the ultimate performance between the two models (5.414 versus 4.193), it is our contention that ESM-2 represents a more appropriate embedding model for the test tasks at hand.

| Model | Performance | Novelty | Original | dist(WT) | Diversity |
|---|---|---|---|---|---|
| Ours - original | 5.414 ± 0.710 | 6.891 | 100 % | 10.417 | 9.102 |
| w/o embedding | 0.850 ± 0.062 | 5.230 | 90% | 7.545 | 1.933 |
| Embedding (ESM-1v) | 3.578 ± 0.422 | 7.762 | 100 % | 11.24 | 14.67 |
| Embedding (AbLang) | 4.193 ± 0.541 | 6.008 | 100 % | 13.94 | 11.56 |

Table 5: **Results of ablation studies in mutation tasks.** The term "w/o embedding" denotes the elimination of the embedding model, thereby enabling the exploration and training of the RL policy in the original sequence space. Two distinct embedding models, namely ESM-1 and Ablang, were established. ESM-1 was trained on UR90, resulting in a lower parameter count than EMS-2. Ablang has received training using the antibody sequences that are present in the OAS database. The findings indicate that ESM-2 exhibits the capacity to generalize across diverse protein sequences across distinct datasets. It is noteworthy that in our experimental analysis, the utilization of Ablang and ESM-2 for embedding purposes exhibits a minimal effect on the expenses associated with training.

**The effectiveness of GVF** As shown in Eq. 5, the effective horizon over which predictions are considered depends on the value of the discount factor $\gamma_z$. From Eq. 3, we can find that for the special case of $\gamma_z = 0$ and any value of $\lambda_z$, we still have $G_t^z = Z_{t+1} = Z_\phi(o_t)$. Thus, the TD-error between prediction and target is equivalent to the intrinsic reward provided by random network (Eq. 1). Therefore, we design an ablation experiment to study the effect of changing the temporal prediction horizon by evaluating GVF with different discount factors. We report the results in Fig. 2(a). The research findings indicate that a $\gamma$ value of $0.7$ is more suitable for the given task and environmental conditions. It should be noted that the parameter $\gamma$ primarily influences the selection of the anticipated time horizon. Based on the findings, it can be inferred that the variation in gamma within a considerable range does not significantly impact the ultimate training effect difference.

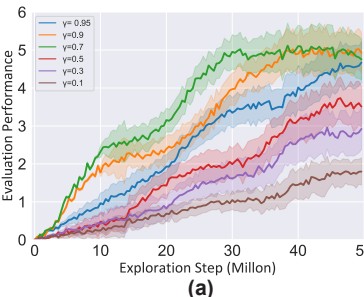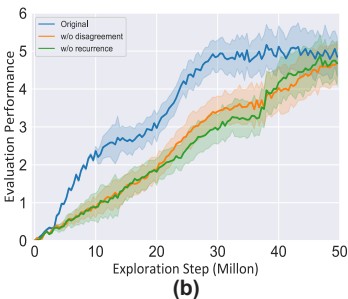

Figure 2: **Results of the abaltion study.** (a) Various $\gamma$ were selected to examine the impact of the anticipated horizon on the ultimate training outcomes. The findings demonstrate that while the vaule of $\gamma$ can impact the ultimate performance outcome, performance sensitivity to $\gamma$ remains relatively low as long as it remains within a reasonable range. (b) A comparison of the impact of each part of the GVF on the final performance. We found that although removing a certain part of disagreeme or recurrence has little effect on final performance, the effect on learning efficiency is very obvious. This finding confirms our hypothesis regarding the influence of the curiosity mechanism on the effectiveness of exploration.

In order to gain a deeper comprehension of the components of GVF in the efficacy of our approach, we have conducted the subsequent ablation analyses: (1) GVF ($\gamma_z = 0$) without the generalized variance term (disagreement) in Eq. 5, (2) GVF ($\gamma_z = 0$) without the recurrent predictor. We report the results of the comparison in Fig. 2(b). The enhanced performance of GVF can be attributed to the integration of the history-conditioned recurrent predictor and the variance term, as evidenced by the results.

# 6 CONCLUSION

In this paper, we introduce a novel protein generation model that learns to generate the protein sequence in a latent space. We further explore an exploration method, curiosity with general value functions, that intrinsically rewards the RL agent based on errors and uncertainty in predicting random transformations of observation sequences generated through the agent's actions. The proposed framework outperformed other baseline methods in performance and is competitive in terms of novelty. The ablation studies prove the necessity of encoding the original sequence space to a latent space, and the effectiveness of the curiosity mechanism. Our approach can be generalized by moving beyond random pseudo-rewards, considering general value functions under a set of different policies. In the future, We will further extend our method in antibody/ protein library design with multi-develop ability.

**Limitations** The function evaluation that provides the extrinsic rewards is one of the critical problems in protein engineering. Currently, a highly effective and precise solution to this issue has yet to be identified. Whilst our approach offers an inherent incentive to stimulate the agent's exploration of the latent space in a more effective manner, thereby mitigating this issue to some degree, its ultimate efficacy is also contingent upon the caliber and precision of the extrinsic reward. Given that protein engineering frequently involves the optimization of multiple objectives, and that the protein sequence and structure in practical scenarios are characterized by greater complexity, it may be necessary to expand the latent space in order to develop general value functions that align with multi-objective optimization. This is a potential avenue for future research.

SOCIETAL IMPACT

The exploration of protein sequence generation has the capacity to bring about a transformative impact across diverse domains and yield substantial advantages for society. The production of innovative protein sequences through this study provides opportunities for the exploration of novel drug discovery and development. The capacity to create and manipulate proteins with precise functionalities has the potential to facilitate the production of more efficacious pharmaceuticals, enhanced therapeutic approaches, and remedies for diverse ailments. The aforementioned progress has the potential to significantly enhance human health and welfare by facilitating the provision of more precise and individualized medical interventions.

Although protein sequence generation research holds significant potential, it is crucial to acknowledge and address any potential risks and concerns. A central issue of paramount importance pertains to the ethical ramifications associated with the field of synthetic biology and the utilization of genetically modified organisms (GMOs). It is imperative to exercise caution and adhere to ethical guidelines and regulatory frameworks that govern the utilization of genetically modified organisms (GMOs) when generating and manipulating protein sequences. To address these concerns and uphold public trust, it is crucial to implement responsible research practices, ensure transparency, and engage with stakeholders.

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
