# OpenReview forum: "Curiosity Driven Protein Sequence Generation via Reinforcement Learning"
_ICLR.cc/2024/Conference — Submitted to ICLR 2024_

### Official Review · Reviewer_3CP8 · 2023-10-31

**Soundness:** 3 good
**Presentation:** 2 fair
**Contribution:** 3 good
**Rating:** 5
**Confidence:** 4

**Summary:**

** Firstly, as a reviewer, I am ignoring the fact that the title in the manuscript were unfortunately not changed by the authors. This does not matter to me when evaluating the paper; I’m sure this is not a big issue since the authors can always add it in later versions **

This paper provides a RL based approach for protein sequence generation, and compares to existing approaches including GFlowNets for protein design. Most importantly, it formulates the protein generation problem as a MDP where the learnt policy learns to generate diverse sequences based on an embedding that is learnt from existing ESM or language models. To generate diversity, the authors propose the use of General Value Functions (GVFs) as a way to induce exploration in the embedding space. Experiments are evaluated on two standard protein generation tasks.

**Strengths:**

1. The most significant strength I think is the ability to turn protein generation as a RL problem. I am aware of few other works that try to do this, but this work is useful due to its ability to be able to use existing ESM based language models to turn sequences into embeddings, and then learn a policy decoder to generate more diverse sequences from the language model embedding space. This is a nice formulation and application of RL in drug discovery space. This work is not the first ones to do so, but there is value in this work, compared to parallel lines of efforts utilizing diffusion or flow matching models for protein structure generation. Most importantly, progress in such works for generating diverse protein sequences can potentially in turn help in the generative AI space of protein structure prediction too.

2. The authors try to make use of the ESM embedding space to learn a policy decoder that can in turn generate more diverse sequences. This is in contrast to directly manipulating the raw sequence space with exploratory actions (since action space is then quite big ~20), and the authors make good use of the fact that ESM based language models have been shown to be successful in generating sequences or structures from the learnt embedding spaces.

3. This work has a nice application of GVFs for curiosity driven exploration. The idea to use GVFs for exploration, and using disagreements from multiple value functions is not new; and exploration literature in RL has exploited this quite a bit - but to be able to explicitly generate diverse samples by using GVFs in drug discovery space is a nice application.

4. The paper is quite nicely written and easy to follow; even for people with perhaps no background in protein generation. The authors do a great job in laying out the context of the application domain and compares to several baselines that are perhaps popular in this space of drug discovery.

5. Experimental results are evaluated for both generation and mutation tasks, and tackles for metrics including novelty and diversity of samples. Results are compared with few baselines, although the draft is missing context of some of these baselines (and there is no supplementary/appendix either)?

**Weaknesses:**

1. On a high level, I think the biggest weakness is that the paper was rushed towards the deadline, and could not do a good job in wrapping up the paper properly. For example, the supplementary is missing; it does include related works section, but this section or experiments section should perhaps contain more details about the baselines considered, what makes the RL approach potentially better than existing ones such as GFNs; or even details on what the DynaPPO approach means in the context of this work?

2. The work tries to use GVFs for inducing diversity in the generated sequences from the embedding space. Since the curiosity is driven based on disagreements from multiple value functions, a natural question is why not use other much simpler baselines for exploration? For example, one can actually do curiosity driven based on an inverse model, or use approaches such as Bootstrapped DQN, which are potentially much easier to implement/run and compare with?

3. I would like to know why the authors chose to use GVFs in this work? What’s the intuition behind using GVFs compared to others?
The authors directly talk about ESM-2 to learn the embeddings from the sequences; for context, ESM-2 would give embeddings, but can it also generate structures and sequences itself? For example, other approaches including AlphaFold, RosettaFold or ESMFold, and there includes several variations of protein folding algorithms now, which can actually generate structures and sequences from the sequences itself? Why did the authors chose to use the pre-trianed ESM-2 model here? What’s the justification?

4. A major potential weakness of the work is that even if it evaluates on metrics including novelty and diversity, if I understand correctly, the proposed approach does not really do better in terms of novelty/diversity? Experimental results always show that performance can be better with the proposed approach? What about the other metrics? I thought the biggest contribution was to be able to generate diverse samples? Are we not achieving that here?

5. I think context of baselines and what they are actually doing should be included in the appendix; for example, what does it mean to have the BO baseline or the DynaPPO baseline? How are these implemented? I would like to know more details about baselines for comparison?

6. What about comparisons with other exploration techniques, or even if we dont use any exploration at all? I see the need for it, but I would like to understand what happens if we just naively use PPO or any other RL algorithm, without the exploration part? How do those results compare with GFNs for example?

7. What’s the task reward or the extrinsic reward here? I agree depending on the task it changes, and these are really sparse reward tasks; but more context about what the extrinsic rewards are would perhaps be helpful.

8. An appendix section including details of the different tasks would be nice; just to see the difficulty of these tasks, and details on baselines, would really help evaluate the novelty/signifciance of the work. To repeat, the application of existing RL methods in these domains is of significance to me already; so I am not questioning that - I am perhaps more worried that are we comparing the baselines correctly, and what is the context of how these baselines are used for comparison?

**Questions:**

Please see the list of questions in the weakness section; other than those, few other questions are :

1. The use of the embedding space learnt from the language model seems to be a key here; to then learn the policy decoder. What’s the intuition or how do we know that the embedding space is good here? Is there any structure that these folding algorithms learn from the sequences that can be useful? How do we know that the decoder policy is indeed generating back diverse sequences without looking at the performance metric?

2. In these tasks, I think information bottleneck based approaches on the embedding space might be quite helpful. I would encourage the authors to explore this too - the existing RL literature has several ways to add bottlenecks on the embedding space. I wonder how the performance would vary if we can add information bottlenecks on top of the pre-trained embeddings, to then learn the policy decoder?
I would really like to see some comparisons with other exploration methods for justifying the use of GVFs for curiosity.

3. The section where the sequences are mapped to an embedding and then the tricks to map back to the sequence space is a bit confusingly written and hard to follow. I would encourage the authors to clear this out properly - since it is one of the key things of the paper + justification of how the ESM-2 embedding space is indeed useful here.

---

### Official Review · Reviewer_J6eo · 2023-11-01

**Soundness:** 1 poor
**Presentation:** 1 poor
**Contribution:** 1 poor
**Rating:** 3
**Confidence:** 4

**Summary:**

A method for protein sequence design with reinforcement learning in the latent space of a protein generative model is introduced. A curiosity mechanism is used to improve search efficiency. The approach is compared against reasonable baselines on two protein sequence generation tasks.

**Strengths:**

- Protein sequence design is a challenging and interesting problem of relevance to the ML community
- The proposed method is compared against a wide variety of performant baselines on multiple tasks

**Weaknesses:**

The writing quality is very poor. Beyond needing a thorough proofreading pass to correct grammar errors and typos, I found it difficult to understand how the method works. Various details about the problem setting, algorithm, and experiment design are missing or poorly described. The paper uses a lot of jargon that could be removed, or at least defined carefully first.

Perhaps due to the unclear presentation of the key ideas, I feel that this work has questionable novelty and significance. Perhaps if I understood the algorithm better, I would have a better sense for the quality of the contributions.

I have provided some questions which may help guide the authors to restructure and revise their manuscript.

I think a simple baseline is missing from the experiments; one could freeze the encoder/decoder networks and iteratively apply a small, random perturbation to the latent vector?
Another baseline for comparison (and for discussion) that appears to be similar to the proposed method is the Deep Manifold Sampler (DMS) [1]

1. Gligorijević, Vladimir, Daniel Berenberg, Stephen Ra, Andrew Watkins, Simon Kelow, Kyunghyun Cho, and Richard Bonneau. "Function-guided protein design by deep manifold sampling." bioRxiv (2021): 2021-12.

**Questions:**

- What are the details of the specific RL formulation assumed by the proposed algorithm? What are the states/observations, actions?
- What is the specific protein design problem under consideration? Is this *de novo* design (generating a sequence of amino acids starting from "scratch" (e.g., random noise)), or directed evolution (applying mutations to a known wild type protein). Or both?
- What form does the policy function take specifically? The paper is confusing about this. Based on Figure 1b, it looks like the RL policy outputs an action in the space of encoded sequence representations... what is an "action" then? A perturbation in this latent space? In the Introduction and Sec. 4.2, the paper mentions that the action space remains "discrete and intricate" and refers to selecting from a pool of N = 20 actions (the number of amino acids?)
- If the curiosity-based exploration is being applied in the latent space of the encoder, then why is it needed? In this case, exploration is "easy" since a small perturbation in this latent space should (in theory) correspond to a large "jump" in sequence space between natural proteins.
- I couldn't follow the descriptions of the part of the algorithm in Sec. 4.1. Please revise and consider including a figure to illustrate?
- How do the methods proposed in the cited papers in Sec. 2 differ from what is presented in this paper?
- Are any of the GVF aspects of this method in Sec 4.2 and 4.3 novel? If not, this can be condensed to allow more space for describing the overall algorithm comprehensively and clearly.


### Experimental setup

- Is the "functionality predictor" the same as the GVF predictor $\hat{v}_{\pi,z}$? Are the ESM2 encoder and decoders fine-tuned or frozen?
- In general, is there a policy function that learns the perturbations applied to the latent encodings (the "actions")? I struggled to follow even basic aspects of the approach such as this...
- What is $f$ in the performance metric? How is the fact that the generated designs are synthetic and thus no real ground truth for protein functionality is available addressed?
- Why are 160 sequences generated and why are only the top 10 evaluated?
- How are the "specific nucleotide positions" selected for Random - 1 and Random - 5 baselines (Table 2)? I would suggest referring to [2] and consider using Simulated Annealing as well as PPDE as strong MCMC baselines for the mutation tasks [2]

[2] Emami, Patrick, Aidan Perreault, Jeffrey Law, David Biagioni, and Peter St John. "Plug & play directed evolution of proteins with gradient-based discrete MCMC." Machine Learning: Science and Technology 4, no. 2 (2023): 025014.

---

### Official Review · Reviewer_yXq2 · 2023-11-01

**Soundness:** 3 good
**Presentation:** 2 fair
**Contribution:** 3 good
**Rating:** 5
**Confidence:** 2

**Summary:**

This paper provides a novel protein generation method based on reinforcement learning.  Specifically, the latent embedding is used for better search; GVF is used to enhance exploration. Experiments reveal that the proposed method outperforms baselines in different generation tasks.

**Strengths:**

1) The proposed method outperforms SOTAs w.r.t performance in different generation tasks.

**Weaknesses:**

1) The authors claim that searching in the embedding space can be interpreted as traversing a nearby functionality or fitness landscape within this representation space. However, there is no experimental evidence provided to support this claim. It would be helpful if the authors could include experiments to demonstrate this point.


2) The proposed method performs worse than the baselines in some metrics (e.g., Novelty in table 1).


3) There are several grammar errors and typos in the paper. Firstly, the title should be "Curiosity Driven Protein Sequence Generation via Reinforcement Learning" instead of "FORMATTING INSTRUCTIONS FOR ICLR 2024 CONFERENCE SUBMISSIONS". Secondly, the "SOCIETAL IMPACT" section should be placed after the references to avoid exceeding the 9-page limit for the main text. Finally, it is advisable to carefully proofread the paper to correct any spelling or grammatical errors throughout the document.

**Questions:**

N/A

---

### Meta-Review · Area_Chair_NP1B · 2023-12-06

**Metareview:**

The paper addresses the challenges in protein sequence design by implementing a reinforcement learning algorithm based on a continuous latent space.

The reviewers have expressed criticism of the paper and highlighted multiple issues. The authors, however, have not provided any feedback or responses.

**Justification For Why Not Higher Score:**

The reviewers have expressed criticism of the paper and highlighted multiple issues. The authors, however, have not provided any feedback or responses.

**Justification For Why Not Lower Score:**

N/A

---

### Decision · Program_Chairs · 2024-01-16

Reject